FTH1 indicates poor prognosis and promotes metastasis in head and neck squamous cell carcinoma

Liao Qingyun 1
Yang Jing 2 3
Lu Zhaoyi 4
Jiang Qingshan 1
Gong Yongqian 1
Liu Lijun 1
Peng Hong 1
Wang Qin 1
Zhang Xin 4
Liu Zhifeng liuzf@usc.edu.cn 1
1 The First Affiliated Hospital, Department of Otolaryngology, Hengyang Medical School, University of South China , Hengyang , Hunan , China
2 Cancer Research Institute, Hunan Province Key Laboratory of Tumor Cellular & Molecular Pathology, University of South China , Hengyang , Hunan , China
3 The First Affiliated Hospital, Department of Gastroenterology, Hengyang Medical School, University of South China , Hengyang , Hunan , China
4 Otolaryngology Major Disease Research, Key Laboratory of Hunan Province, Central South University , Changsha , Hunan , China
Patnaik Santosh
Electronic publication date: 2023 Nov 17
Publication date: 2023
Volume: 11
Electronic Location ID: e16493
Received 2023 Jul 3; Accepted 2023 Oct 30
Copyright: ©2023 Liao et al.
Copyright year: 2023
Copyright holder: Liao et al.
License: This is an open access article distributed under the terms of the Creative Commons Attribution License, which permits unrestricted use, distribution, reproduction and adaptation in any medium and for any purpose provided that it is properly attributed. For attribution, the original author(s), title, publication source (PeerJ) and either DOI or URL of the article must be cited.
License URL: https://creativecommons.org/licenses/by/4.0/

Keywords: HNSCC (head and neck squamous cell carcinoma), FTH1 (ferritin heavy chain), Prognosis, Metastasis, Biomarker

Funding: The Hunan provincial Health and Family Planning Commission B202307019501 202103030223 20201947 B20180186 The Hunan Provincial Natural Science Foundation of China 2021JJ40502 2019JJ50547 This study was supported by grants from the Hunan provincial Health and Family Planning Commission (B202307019501, 202103030223, 20201947, B20180186), the Hunan Provincial Natural Science Foundation of China (2021JJ40502, 2019JJ50547). The funders had no role in study design, data collection and analysis, decision to publish, or preparation of the manuscript.

==============================
Background

Currently, ferritin heavy chain (FTH1) has been increasingly found to play a crucial role in cancer as a core regulator of ferroptosis, while its role of non-ferroptosis in head and neck squamous cell carcinoma (HNSCC) is still unclear.

Methods

Herein, we analyzed the expression level of FTH1 in HNSCC using TCGA database, and FTH1 protein in HNSCC tissues and cell lines was determined by immunohistochemistry (IHC) and western blotting, respectively. Then, its prognostic value and relationship with clinical parameters were investigated in HNSCC patients. Additionally, the biological function of FTH1 in HNSCC was explored.

Results

The current study showed that FTH1 is significantly overexpressed in HNSCC tissues and related to poor prognosis and lymph node metastasis of HNSCC. FTH1 knockdown could suppress the metastasis and epithelial-mesenchymal transition (EMT) process of HNSCC.

Conclusion

Our findings indicate that FTH1 plays a critical role in the progression and metastasis of HNSCC and can serve as a promising prognostic factor and therapeutic target in HNSCC.

Introduction

Head and neck cancers are the sixth most common tumor globally, arising in the upper respiratory tract, of which squamous cell carcinoma accounts for about 90% (Bray et al., 2018). The pathogenesis of the incidence of HNSCC between Occident and China is very different. The leading causes of HNSCC in China are smoking and drinking, while the incidence of HNSCC caused by HPV infection has increased, especially oropharyngeal squamous cell carcinoma in the Occident (Chaturvedi et al., 2011). The incidence of HNSCC is increasing globally, but the continuous development of traditional surgery, radiotherapy, chemotherapy, targeted therapy and even immunotherapy have failed to improve the 5-year survival rate of patients significantly. Compared with HPV-positive patients, the treatment prognosis of negative patients is relatively poor, which prompts scientists to seek personalized treatment based on individual patients’ genomic information (Johnson et al., 2020; The Cancer Genome Atlas Network, 2015). An in-depth understanding of the molecular biological mechanisms of HNSCC will help discover new molecular biomarkers and potential targets, also provide a basis for early diagnosis and precise treatment.

In 1937, Laufberger isolated a new protein from horse spleen with iron content as high as 23% of dry weight, called ferritin (Laufberger, 1937). Although there were some reports of ferritin found in human serum, it was not until 1972 that Addison used immunoradiometric assay to confirm that ferritin does exist in human serum (Addison et al., 1972). Ferritin is composed of 24 middle-hollow round subunits, including heavy chain (FTH1) and light chain (FTL1) (Arenas-Salinas et al., 2014; Goralska et al., 2005). Ferritin plays essential functions such as iron metabolism, signal transduction, immunity, angiogenesis and inflammation, and plays a vital role in many diseases and even tumors (Hazard & Drysdale, 1977; Wang et al., 2010). Dysregulated iron metabolism is a tumor Hallmark (Torti & Torti, 2013). In HNSCC, it was first reported by Maxim & Veltri (1986) that ferritin may be a valuable tumor biomarker. In the serum of HNSCC patients, the ferritin content was significantly higher than in the healthy group and the ferritin content in advanced patients was also much higher than that in early-stage patients. There is evidence that the ferritin content in HNSCC patients decreased significantly after receiving successful treatment for five months and returned to normal after five years (Hu et al., 2019). Among the genes encoding the light and heavy chain proteins in ferritin, only FTH1 has ferroxidase activity, while FTL1 is mainly related to iron nucleation and the stability of assembled ferritin (Arosio & Levi, 2002). Recent proteomics studies of some tumors have found that ferritin is mainly present in the form of ferritin H in malignant tissues (Lukina et al., 1993). In recent years, FTH1 as a master regulator of ferroptosis has been reported as a prognostic factor in brain cancer, pancreatic cancer, kidney cancer and breast cancer (Chekhun et al., 2014; Huang et al., 2019; Rosager et al., 2017; Su, Lei & Zhang, 2017). However, the prognostic value and biological function of FTH1 in HNSCC is still worth further exploration.

Herein, we aimed to clarify the clinical significance and biological function of FTH1 in HNSCC. The results reveal that the expression of FTH1 in cancer was elevated compared to adjacent tissues and FTH1 overexpression indicated a higher risk of lymph node metastasis and poor prognosis of patients with HNSCC. Furthermore, FTH1 deletion dramatically inhibited the metastasis and EMT of HNSCC cancer cells. Collectively, the current study suggests that FTH1 represented a novel prognostic and metastatic biomarker and a potential therapeutic target for HNSCC. A preprint has previously been published (Liu et al., 2022).

Materials & methods

Data acquisition and tissue specimens

A workflow framework of this research is presented in Fig. 1. The RNA-Seq and clinical data of HNSCC dataset (528 cases) were downloaded from The Cancer Genome Atlas (TCGA) database. A total of 499 HNSCC samples, with 41 cases of matched adjacent normal tissues, without missing expression and missing follow-up were selected for the subsequent analysis. Besides, 21 sets of HNSCC tissue sections including cancer and adjacent normal tissues used for Immunohistochemistry (IHC) were collected from the First Affiliated Hospital of the University of South China. None of the patients received any preoperative anticancer treatment before surgical procedures. Their pathological diagnosis was confirmed by at least two pathologists. This study was approved by the Medical Ethics Committee of The First Affiliated Hospital of University of South China (Number: 2021ll0916001) and the requirement of consent was waived for the retrospective analysis.

Figure 1 The workflow framework of this study.

Immunohistochemistry (IHC)

IHC was performed as our previous studies (Deng et al., 2020; Gong et al., 2018), using the PV-9000 IHC Reagent (ZSGB-BIO, Beijing, China). In brief, tissue sections were dewaxed with turpentine and then hydrated with a gradient of decreasing concentration of ethanol. The antigen retrieval was executed by boiling 10 mmol/l citric acid buffer (pH 6.0) for 15 min. After endogenous peroxidase was inactivated, the sections were blocked with normal goat serum (ZSGB-BIO, Beijing, China) for 10 min. Then the sections were incubated overnight with anti-FTH1 antibody (Affinity; 1:200) at 4 °C, followed by the horseradish peroxidase labeled secondary antibody. For the negative control, the normal rabbit IgG was applied. Finally, the positive signals were visualized by chromogenic agent DAB and nuclei were counter-stained with hematoxylin. Signal strength was scored as follows: 0 (negative), 1 (weak), 2 (moderate) and 3 (strong). The staining distribution score is based on the percentage of positive cells: 0 (0–5%), 1 (5–24%), 2 (25–49%), 3 (50–74%) and 4 (75–100%).

Cell culture and transfection

HNSCC cell lines (Fadu, SCC4, Cal27, HN8 and Cal33), 293T and immortalized non-malignant cell line DOK were obtained from Shanghai Cell Bank (The Chinese Academy of Science, Shanghai) or ATCC. DOK cells were cultured with RPMI 1640 (Gibco). HNSCC cell lines and 293T cells were cultured in DMEM (Gibco, Waltham, MA, USA). The culture mediums were supplemented with 10% FBS (Gibco) and all cells were cultured at 37 °C with 5% CO2.

The shRNA-encoding lentiviral vectors for FTH1 knockdown were purchased from GeneCopoeia (GeneCopoeia, Rockville, MD, USA). The shRNA target sequences for human FTH1 were as follows: shFTH1-1: 5′-CCATGTCTTACTACTTTGACC-3′; shFTH1-2: 5′-CCATCAAAGAATTGGGTGACC-3′. The procedures of lentivirus packaging were performed using Lenti-Pac™ HIV lentivirus packaging kit (GeneCopoeia, Rockville, MD, USA). The negative control, nominated as Lv-control, was packaged with the empty vector. After 48–72 h of transfection, lentiviral particles were harvested to infect cell lines. Stable transfected cells were selected with puromycin (GeneCopoeia; 2 µg/ml) for 2 weeks and the inhibitory efficiency of FTH1 was confirmed by Western blotting.

Cell migration and invasion

Cell migration and invasion assay was conducted using wound healing and Transwell invasion assays as reported in our previous studies (Liu et al., 2020; Yang et al., 2020). For wound healing assay, stable transfected cells were seeded in a 6-well plate and scratched using a 10 µl tip when cells reached 90% confluency and then cells were cultured using serum-free medium for 48 h. Transwell invasion assays were performed using Transwell chambers (Corning, NY, USA) pre-coated with 15% Matrigel (Corning, USA). Cells were seeded into the upper chamber (2  × 104 cells/well), while the lower chamber was placed with the medium containing 10% FBS. After 48 h, the cells on the lower surface were fixed utilizing paraformaldehyde and stained using crystal violet. After removing the cells on the upper surfaces, the stained cells were counted under a microscope.

Gene set enrichment analysis (GSEA)

Gene set enrichment analysis (GSEA) was performed using GSEA 4.0.1 software to explore the potential functions enriched in subgroups of high or low expression of FTH1. The gene set ‘c2.cp.kegg.v7.4. symbols.gmt [curated]’ from the MSigDB database was used as reference for GSEA. The enriched sets with P-value < 0.05 and FDR (false discovery rate) < 0.25 were considered statistically significant.

Western blotting

Whole-cell proteins were extracted using RIPA lysis buffer (NCM Biotech, China) with the proteasome inhibitor (Beyotime Biotechnology, Jiangsu, China) and centrifugated to collect the supernatant. After determination of the protein concentration, the collected supernatant was added with SDS buffer, incubated for 10 min at 100 °C, separated by SDS-PAGE and then transferred to the PVDF membrane (Millipore, Bedford, MA, USA). The membranes blocked with 5% skimmed milk was incubated overnight with primary antibody against FTH1 (1:1000 dilution; Affinity), E-cadherin (1:2000 dilution; Proteintech), N-cadherin (1:1000 dilution; CST), Vimentin (1:1000 dilution; CST) at 4 °C. Finally, the antigen-antibody complexes were visualized using enhanced chemiluminescence reagents (Thermo Fisher Scientific).

Statistical analysis

All statistical analyses were performed using R software (version 4.1.0; R Core Team, 2021). Student’s t test and Wilcox test were used to evaluate differences between two groups. Kruskal Wallis test and one-way analysis were used to analyze differences among multiple groups. Overall survival (OS) and disease-free survival (DFS) analysis were conducted using the Kaplan–Meier method with the log-rank test. The chi-square test and Fisher’s exact test were utilized to analyze the correlation between FTH1 expression and the clinicopathologic parameters. Univariate and multivariate Cox regression analyses were performed to identify independent prognostic factors. A value of P < 0.05 indicates statistical significance. Asterisks (*, **, ***) represented P < 0.05, P < 0.01 and P < 0.001, respectively.

Results

Differentially expression and prognostic value of FTH1 and FTL1 in HNSCC

The differential analysis of TCGA-HNSC transcriptome data showed that the expression levels of FTH1 and FTL1 in HNSCC samples were significantly higher than those in adjacent normal samples (Figs. 2A, S1A), which were also confirmed using paired sample analysis (Figs. 2B, S1B). FTH1 in HNSCC cells was overexpressed compared with DOK, an immortalized non-malignant cell (Fig. 2C). The IHC staining showed that FTH1 protein was highly expressed in HNSCC tissues compared with matched adjacent epithelial tissues (Figs. 2D–2F). Based on FTH1 and FTL1, the coding genes for the heavy chain and light chain of ferritin significantly highly expressed in tumor samples from HNSCC patients, and we further analyzed their clinical prognostic value. In the Kaplan–Meier survival estimate, we found that only high expression of FTH1 in tumors indicates worse OS probability (P = 1.39e−03) and DFS probability (P = 4.054e−05), while FTL1 has no prognostic value (Figs. 3A and 3B, Figs. S1C and S1D). From the above results, it can be indicated that the expression of FTH1 has the prognostic value of OS and DFS simultaneously and the clinical outcomes of patients with high FTH1 expression are worse.

Figure 2 Elevated expression of FTH1 in HNSCC tissues.

(A) Differential expression of FTH1 in HNSCC and adjacent normal tissues in TCGA dataset. (B) Paired difference analysis of FTH1 mRNA expression in TCGA-HNSCC dataset. (C) The protein level of FTH1 in HNSCC cell lines and immortalized non-malignant cell line DOK was tested by western blot. Representative IHC staining demonstrates the expression of the FTH1 protein in adjacent normal (D) and HNSCC (E) tissues (100 µm: 200×, 50 µm: 400×). (F) The IHC score was significantly higher in HNSCC tissues than in adjacent tissues. *** P < 0.001.

Figure 3 High levels of FTH1 predicts a worse prognosis of HNSCC patients.

Kaplan–Meier curves showed OS (A) and FDS (B) in HNSCC patients in terms of FTH1 expression.

Correlation between FTH1 and clinical parameters

We generalized these patients’ baseline parameters with age, gender, smoking, drinking history, pathological grade, clinical stage, T/N/M stage, OS and DFS according to the high and low expression levels of FTH1 (Table 1). The preliminary correlation between FTH1 expression and clinicopathological features was obtained by logical regression analysis, as shown in Table 2. It shows that FTH1 expression is associated with smoking (no vs. yes, P = 0.008), pathological grade (G3 & $ vs. G1 & @, P < 0.01), N stage (N+ vs. N0, P < 0.01) and clinical stage (Stage III & IV vs. Stage I & II, P  < 0.05). After that, we further performed a Wilcoxon signed-rank test on clinicopathological features that distribute in high and low FTH1 expression groups. In addition to the differences in FTH1 expression among patients with a history of drinking (P < 0.001), the results of smoking (no vs. yes, P < 0.05), pathological grading (G3 & $ vs. G1 & 2, P < 0.01), N staging (N+ vs. N0, P < 0.01) and clinical stage (Stage III & IV vs. Stage I & II, P < 0.05) conform to logistic regression (Figs. 4A–4I).

Table 1 Clinicopathologic parameters of TCGA HNSCC patients.

Clinicopathologic parameters	FTH1 expression	P-value	
	Low (n = 250)		High (n = 249)		
Age						0.177	
<60	118	23.65%		102	20.44%		
≥60	132	26.45%		147	29.46%		
Gender						0.418	
Female	71	14.23%		62	12.42%		
Male	179	35.87%		187	37.47%		
Smoker						0.009	
No	68	13.63%		43	8.62%		
Yes	177	35.47%		201	40.28%		
Alcohol history						0.059	
No	34	6.81%		21	4.21%		
Yes	69	13.83%		80	16.03%		
Histologic grade						0.001	
G1 + G2	196	39.28%		163	32.67%		
G3 + G4	45	9.02%		76	15.23%		
T stage						0.186	
T1 + T2	94	18.84%		81	16.23%		
T3 + T4	146	29.26%		163	32.67%		
N stage						0.001	
N0	99	19.84%		140	28.06%		
N1-N3	134	26.85%		104	20.84%		
M stage						0.372	
M0	234	46.89%		235	47.09%		
M1	1	0.20%		4	0.80%		
Clinical stage						0.018	
Stage I + Stage II	67	13.43%		46	9.22%		
Stage III + Stage IV	173	34.67%		199	39.88%		
OS event						<0.001	
Alive	186	37.27%		147	29.46%		
Dead	64	12.83%		102	20.44%		
DFS event						<0.001	
Free	184	36.87%		132	26.45%		
No	52	10.42%		86	17.23%		

Table 2 Correlation between the clinicopathologic parameters and FTH1 expression (logistic regression).

Parameters	Total (n)	Odds ratio (OR)	P-value	
Age (>60 vs.≤60)	499	1.288 (0.904–1.838)	0.161	
Gender (Male vs. Female)	499	1.196 (0.804–1.783)	0.377	
Smoker (Yes vs. No)	489	1.796 (1.17–2.781)	0.008	
Alcohol history (Yes vs. No)	489	1.877 (1.004–3.573)	0.051	
Histologic grade (G3+4 vs. G1+2)	480	2.031 (1.335–3.117)	0.001	
T stage (T3+4 vs. T1+2)	484	1.296 (0.894–1.881)	0.172	
N stage (N+ vs. N0)	477	1.822 (1.269–2.625)	0.001	
M stage (M1 vs. M0)	474	3.983 (0.584–78.228)	0.218	
Clinical stage (Stage III + IV vs. Stage I + II)	485	1.675 (1.096–2.579)	0.018	

Figure 4 Correlation between FTH1 expression and clinicopathologic parameters.

Distribution of the FTH1 expression stratified by clinicopathologic parameters: (A) pathological grade (G1 & 2 and G3 & 4), (B) T stage (T1 & 2 and T3 & 4), (C) N stage (N0 and N+), (D) M stage (M0 and M1), (E) clinical stage (Stage I & II and Stage III & IV), (F) Age (≤60 and >60), (G) Gender (Female and Male), (H) smoking (No and Yes), (I) Alcohohistory (No and Yes). ns P > 0.05; * P < 0.05; ** P < 0.01; *** P < 0.001.

We further performed a stratified Kaplan–Meier analysis for OS and DFS in TCGA-HNSC patients with the differential clinical features related to FTH1 expression. The related results showed that FTH1 expression had the prognostic value of OS (P = 0.004) and DFS (P < 0.001) in the G1 & 2 stage, but not in the G3 & 4 group (Figs. 5A and 5B, Figs. 6A and 6B). FTH1 expression has prognostic value of OS (both P < 0.05) and DFS (both P < 0.05) whether lymphatic metastasis or not (Figs. 5C and 5D, Figs. 6C and 6D); FTH1 expression has a predictive value of DFS (P = 0.002) but no predictive value of OS (P = 0.072) in the early stage (Stage I & II) population, while in advanced patients, have prognostic value regardless of OS (P = 0.013) or DFS (P = 0.004) (Figs. 5E and 5F, Figs. 6E and 6F). Finally, we were pleasantly surprised to find that FTH1 expression has a good prognostic value for both OS and DFS in smokers (P < 0.001), while not in non-smokers (Figs. 5G and 5H, Figs. 6G and 6H). From the results of the stratified analysis, we guess that FTH1 expression has better prognostic value for smokers, advanced and pathologically well-differentiated patients, and its prognostic value is not affected by the status of lymph node metastasis.

Independent predictive power of FTH1

The Cox regression analysis results of FTH1 expression level and clinicopathological features for OS prognosis have been shown in Table 3. Univariate cox analysis showed that metastasis (P = 0.026, HR: 3.721, 95% CI [1.177–11.764]) and FTH1 expression (P < 0.001, HR: 1.646, 95% CI [1.254–2.161]) were statistically significant. Moreover, multivariate cox analysis showed that both of them still had good statistical significance (metastasis, P = 0.031, HR: 3.547, 95% CI [1.122–11.214]) and (FTH1 expression level, P < 0.001, HR: 1.658, 95% CI [1.252–2.194]). The above analysis results indicate that in addition to the independent prognostic value of metastasis, FTH1 expression level can also be used as an excellent independent prognostic factor.

GSEA reveal FTH1-related pathways and molecular functions

Through GSEA, we explored the differences in the downstream activated signaling pathways between the low and high FTH1 expression groups to search for its potential carcinogenic mechanism. After screening of high expression in the group in MSigDB gene set (c2. Cp. Kegg. V7.1. Symbols. gmt), we found that 24 significant enrichment pathways (FDR < 0.25, NOM p-value < 0.05) are mainly concentrated in six aspects between the degree of enrichment of pathways: energy metabolism, glycometabolism, protein and amino metabolism, other metabolisms, cell adhesion and motility and tumor-associated signal pathways (Figs. 7A–7D).

Figure 5 Stratified Kaplan–Meier analysis for OS in HNSCC patients.

Stratified Kaplan–Meier analysis for OS with the differential clinical features, (A) Grade 1+2, (B) Grade 3+4, (C) N0, (D) N+, (E) Stage I+II, (F) Stage III+IV, (G) smoking, (H) no smoking, related to FTH1 expression in HNSCC patients.

Figure 6 Stratified Kaplan–Meier analysis for DFS in HNSCC patients.

Stratified Kaplan–Meier analysis for DFS with the differential clinical features, (A) Grade 1+2, (B) Grade 3+4, (C) N0, (D) N+, (E) Stage I+II, (F) Stage III+IV, (G) smoking, (H) no smoking, related to FTH1 expression in HNSCC patients.

Table 3 Univariate and multivariate cox regression of OS and clinicopathologic parameters in HNSCC patients.

Parameters	Univariate analysis		Multivariate analysis	
	HR (95% CI)	P-value		HR (95% CI)	P-value	
Age (>60 vs.≤60)	1.252 (0.956–1.639)	0.102				
Gender (Male vs. Female)	0.764 (0.574–1.018)	0.066				
Smoker (Yes vs. No)	1.089 (0.778–1.525)	0.618				
Alcohol history (Yes vs. No)	0.952 (0.716–1.265)	0.733				
Grade (G3+4 vs. G1+2)	0.939 (0.688–1.282)	0.692				
T stage (T3+4 vs. T1+2)	1.245 (0.932–1.661)	0.137				
N stage (N+ vs. N0)	1.263 (0.964–1.653)	0.090				
M stage (M1 vs. M0)	3.721 (1.177–11.764)	0.026		3.547 (1.122–11.214)	0.031	
Clinical stage (III+IV vs. I+II)	1.217 (0.878–1.688)	0.238				
FTH1 (High vs. Low)	1.646 (1.254–2.161)	<0.001		1.658 (1.252–2.194)	<0.001	

Figure 7 Enrichment results from multiple GSEA.

(A) Energy metabolism, (B) glycometabolism, (C) protein and amino metabolism, (D) other metabolisms, (E) cell adhesion and (F) motility and tumor-associated signal pathways.

FTH1 knockdown suppresses HNSCC metastasis by attenuating EMT

As previously mentioned, FTH1 was associated with lymph node metastasis. Hence, wound-healing and Transwell assays were performed to identify FTH1 affecting migration and invasion capabilities of HNSCC cells. Attractively, FTH1 knockdown suppressed wound healing rates in Fadu and HN8 cells (Figs. 8A–8D). Similar results were obtained in the Transwell invasion assays (Figs. 8E and 8F). Furthermore, we investigated molecular markers of EMT by Western blotting. After FTH1 depletion, mesenchymal markers (N-cadherin and vimentin) are suppressed, while the epithelial marker E-cadherin was upregulated (Figs. 9A and 9B).

Figure 8 FTH1 inhibition suppressed HNSCC cell migration.

Migration ability of HNSCC cells (A–D) were tested and quantified by wound healing assay. Invasive ability was examined and quantified by Transwell invasion assay (E and F).

Figure 9 FTH1 knockdown suppresses HNSCC EMT.

(A and B) Western blotting was performed to examine the expression of E-cadherin, N-cadherin and vimentin proteins after FTH1 depletion.

Discussion

Currently, FTH1 has attracted much attention as a core regulator of ferroptosis (Tang et al., 2021), but its role of non-ferroptosis in HNSCC is still ambiguous. The current study suggests that the role of FTH1 in tumors depends on the context in which it is present. FTH1 can be used as a tumor promoter in metastatic melanoma cells (Di Sanzo et al., 2011a), brain cancer (Rosager et al., 2017), pancreatic cancer (Su, Lei & Zhang, 2017) and a tumor suppressor in non-small cell lung cancer (Biamonte et al., 2018a) and ovarian cancer (Lobello et al., 2016), while the role of FTH1 in breast cancer is still controversial (Aversa et al., 2017b; Chekhun et al., 2014). In this study, we discovered that FTH1 is significantly overexpressed in HNSCC tissues in the TCGA-HNSC database, and further, IHC was used to verify the above results. Simultaneously, we found that the high expression of FTH1 correlated with lymph node metastasis and higher pathological grade and clinical stage can act as an independent predictor of the poor prognosis of HNSCC. The results are consistent with previous studies of FTH1 in multiple solid tumors, except for clinical stage (Ali et al., 2021; Hu et al., 2021a; Hu et al., 2021b; Huang et al., 2019). Hence, we aimed to the biological function of FTH1 in HNSCC, and the results demonstrated that endorsed the metastasis of HNSCC cells. Briefly, the current study indicates that FTH1 plays a vital role in the pro-oncogenic functions and is a potential biomarker and therapeutic target in HNSCC.

Iron metabolism plays a crucial role in cancer metastasis by affecting some enzymes activities. Iron overload increases the activity of metalloprotease-2/9 (MMP-2/9) activating AP-1 via the ERK/Akt pathway (Kaomongkolgit et al., 2008). Increased iron concentration caused by FPN overexpression attenuates the ROS generation and impedes EMT (Shan, Wei & Shaikh, 2018). The role of ferritin as an important iron metabolism regulator has been extensively studied in tumors. However, there are few studies focusing on ferritin subunits FTH1 and FTL1. Among them, only FTH1 has the enzyme activity to oxidize divalent iron to trivalent iron (Timoshnikov et al., 2015). FTH1 helps the synthesis of ferritin by enhancing the storage of iron in cells and mainly affects tumors’ progress by regulating iron metabolism. The ratio of FTH1 and FTL1 in ferritin is specific, and ratio H/L and ferritin are essential for cell survival. These two subunits are not interchangeable, and FTL1 cannot compensate for the function of FTH1 (Ferreira et al., 2001). FTH1 plays an essential role in the regulation of proliferation, angiogenesis, migration, EMT, stemness, inflammation and immunoregulation. At present, the research of FTH1 in tumor pathogenesis mainly exerts its functions by affecting iron metabolism. In the study of the proteomics in human metastatic melanoma cells knocking down FTH1, 200 differential proteins were found, mainly distributed in metabolic pathways related to tumor progression and metastasis (Di Sanzo et al., 2011b). Knockdown FTH1 can inhibit the growth and invasion of melanoma. The regulation of iron depletion by FTH1 can slow down the self-renewal of breast cancer stem cells (Kanojia et al., 2012), while silencing FTH1 in SKOV3 cells can promote the stemness of cervical cancer cells and the up-regulation of NANOG, SOX2, OCT4 (Lobello et al., 2016). After down-regulating FTH1 in breast cancer, cervical cancer and non-small cell lung cancer, cancer cells tend to be in a mesenchyme state, which helps tumor metastasis (Aversa et al., 2017a; Lobello et al., 2016). Here, we found that FTH1 silencing hampers the EMT process in HNSCC cells. Our results illuminate that FTH1 functions as a crucial regulator of EMT to enhance the metastasis of HNSCC.

In addition to regulating iron metabolism, FTH1 also directly acts on oncogenes, oncomiRs and the chemokine pathway. Dongiovanni et al. (2010) found that iron depletion can up-regulate p53 and induce apoptosis. Further studies in NSCLC found that FTH1 regulates miR-125b/p53 axis up-regulates the pro-apoptotic protein BAX down-regulates anti-apoptotic protein Bcl2 by destroying the mitochondrial membrane potential (MMP) and mediating a cascade of enzymatic apoptosis (Biamonte et al., 2018b). Silenced-FTH1 MCF-7 and H460 cells produced a large amount of ROS and activated the CXCR4/CXCL12 signaling pathway, thereby promoting cancer high migration (Aversa et al., 2017a). Our GSEA analysis results suggest that FTH1 may plays an important role in multiple cell adhesion signal pathways, such us ECM receptor interaction, focal adhesion, gap junction and regulation of actin cytoskeleton.

FTH1 is present not only as a biological marker but can also be used in magnetic resonance imaging (MRI) and nanomaterials. In breast cancer, serum biomarkers (CA 15-3) combined with tumor-associated antigens and autoantibodies (heterogeneous nuclear ribonucleoproteins F and FTH1) can improve the accuracy of breast cancer diagnosis (Dong et al., 2013). FTH1 was used as an MRI reporter gene in liver cancer, which can be used for the diagnosis and treatment of early liver cancer; also, with additional iron added, it was more efficient and safer in nasopharyngeal carcinoma (Feng et al., 2012; Genove et al., 2005; Zhou et al., 2020). FTH1 nanoparticles coated with EGF have been successfully applied to breast cancer in vitro and in vivo (Li et al., 2012). The nano-ferritin-HFt-MP-PAS40-Dox packaged with doxorubicin was safely applied to HNSCC, which had a higher maximum tolerated dose (MTD) and better efficiency (Damiani et al., 2017). Currently, siRNA, miRNA, piRNAs and other carriers can be effective tools for targeted FTH1 in the tumor (Balaratnam, West & Basu, 2018), of which H-ferritin siRNA has achieved initial success in improving the curative effect of gliomas in patients receiving chemotherapy (Liu et al., 2011). Although ferritin is present in serum but not synthesized in serum, it is mainly leaked by damaged tumor cells, and tumor cell damage mainly occurs in the advanced stage (Kell & Pretorius, 2014), so this may cause the ferritin in serum to be unable to predict early HNSCC. The overexpression of FTH1 in tumor tissues may be a good indicator of its cancer-promoting function. In addition to ferroptosis, upregulation of FTH1 may promote the invasion and metastasis of HNSCC. Therefore, the prognostic value of FTH1 for patients in the advanced stages and even posttreatment is worthy of further attention.

However, some limitations of our research should be acknowledged. Firstly, a larger cohort of patients are required for exploring clinical significance. Additionally, the heterogeneity and subsites of HNSCC deserves further exploration. Furthermore, in vivo experiments and validation of intermolecular regulation need to be further completed. Thus, future studies with a larger cohort, in vivo experiments in mice and molecular mechanism are warranted to further validate the current results.

Conclusions

This study found that the high expression of FTH1, not FTL1, could be an independent predictor of the prognosis of HNSCC. In addition, FTH1 downregulation could weaken the metastasis and EMT process of HNSCC. Henceforth, FTH1 could represent a promising biomarker and have value as a therapeutic target for the inhibition of metastasis in HNSCC.

Supplemental Information

Supplemental Information 1 The expression and prognostic value of FTL in HNSCC

(A) Differential expression of FTL in HNSCC and adjacent normal tissues in TCGA dataset. (B) Paired difference analysis of FTL mRNA expression in TCGA-HNSCC dataset. Kaplan–Meier curves showed OS (C) and FDS (D) in HNSCC patients in terms of FTL expression.

Click here for additional data file.

Supplemental Information 2 Raw data

Click here for additional data file.

Supplemental Information 3 Uncropped gels/blots (Figs. 9 & 10)

Click here for additional data file.

Additional Information and Declarations

Competing Interests

Author Contributions

Data Availability

The authors declare there are no competing interests.

Qingyun Liao performed the experiments, prepared figures and/or tables, and approved the final draft.

Jing Yang performed the experiments, authored or reviewed drafts of the article, and approved the final draft.

Zhaoyi Lu analyzed the data, prepared figures and/or tables, and approved the final draft.

Qingshan Jiang analyzed the data, authored or reviewed drafts of the article, and approved the final draft.

Yongqian Gong performed the experiments, prepared figures and/or tables, and approved the final draft.

Lijun Liu performed the experiments, prepared figures and/or tables, and approved the final draft.

Hong Peng analyzed the data, prepared figures and/or tables, and approved the final draft.

Qin Wang performed the experiments, prepared figures and/or tables, and approved the final draft.

Xin Zhang conceived and designed the experiments, authored or reviewed drafts of the article, and approved the final draft.

Zhifeng Liu conceived and designed the experiments, performed the experiments, authored or reviewed drafts of the article, and approved the final draft.

The following information was supplied regarding data availability:

The raw data is available in the Supplemental Files. The initial FTH1 data is available at the TCGA: https://portal.gdc.cancer.gov/.

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
