# Peer review of "FTH1 indicates poor prognosis and promotes metastasis in head and neck squamous cell carcinoma"

_PeerJ, doi:10.7717/peerj.16493_

## Round 0.1 · original submission · Major Revisions

Thank you for submitting this manuscript. We have now obtained two reviews that are appended to this email. Please address both sets of reviewers' concerns prior to re-submission.

·

Basic reporting

In the Introduction Section, line 67-68, “However, the prognostic value ... ... is still unclear.” This description is too absolute. To my knowledge, at least two published articles have contributed to the role of FTH1 in HNSCC, including the relationship between FTH1 expression and the prognosis of patients with HNSCC.
1.Hu ZW, Chen L, Ma RQ. Comprehensive analysis of ferritin subunits expression and positive correlations with tumor-associated macrophages and T regulatory cells infiltration in most solid tumors. Aging (Albany NY) 2021;13(8): 11491-506.
2.Hu ZW, Wen YH, Ma RQ. Ferroptosis driver SOCS1 and suppressor FTH1 independently correlate with M1 and M2 macrophage infiltration in head and neck squamous cell carcinoma. Front Cell Dev Biol 2021;9:727762.
Authors cannot ignore the contributions of the previously published studies in this regard, and these previous work need to be included and discussed in the manuscript.

Experimental design

no comment

Validity of the findings

1.In the survival analysis, the number of clinical cohort need to be clearly addressed. i.e. the case number need to be included in the Results Section 3.1, 3.2, and Figure Legends of Fig 3, 4, 5, 6.

2.In Result Section, line 230-231, The conclusion of “According to the above... ...through HMOX1.” is too absolute, as the interference of HMOX1 was not included in the study. Hence, the author need to perform extra experiment to support their conclusion or results 3.6 should be deleted.

Additional comments

The limitations of this study need to be included in the Discussion Section.

·

Basic reporting

The article is structured in a clear and concise manner with results that are directly relevant to the hypothesis stated. The figures and tables are well organized, and the addition of the workflow in Figure 1 is a helpful addition. However, there are a few minor adjustments that this paper could benefit from. Overall, the article is well written, however special attention could be given to punctuation throughout the article, specifically the use of commas. Be sure to explain any abbreviations used throughout the paper, including in the abstract. The authors could elaborate on TNM staging (tumor, node, metastasis) as it is not well introduced. The introduction would benefit from expanding the background of recent and relevant findings on ferritin, FTH1, and HNSCC and including more updated references/context. Other studies have also found that FTH1 expression is high in HNSCC, was associated with poor OS, and may contribute to invasiveness; it would be beneficial to reference these in your discussion. It may also be beneficial to expand on the role of iron metabolism, generation of ROS, and migration in your discussion. In figure legend 2, E needs to be changed to F when describing the IHC score.

Experimental design

The examination into the effect of FTH1 expression on clinical parameters and the potential role in invasion is well defined. As others have also found that FTH1 expression in HNSCC is associated with poor outcomes, this work is meaningful in further justifying FTH1 as a potential biomarker. It would be helpful to expand on materials and methods use in this research. This includes adding more detail such as use of GSEA and the Molecular Signatures Database, use of protein-protein interaction network, and composition of RIPA buffer. The work investigating FTH1 in relation to invasion could be further supported by additional discussion of findings of enriched pathways from GSEA, specifically cell adhesion. It would be beneficial to expand these results to show that alterations in these pathways in the high FTH1 expression groups supported what was found with the protein expression in Figure 9 or Figure 10.

Validity of the findings

Other studies have also found that FTH1 expression is high in HNSCC, was associated with poor OS, and examined correlation to clinical parameters. This study includes meaningful replication that further supports these findings. One such study (PMID: 34527677) found no significance in relation to clinical stage, differing from your findings-may want to comment on why findings may differ. This article concludes that FTH1 may represent a biomarker in HNSCC that contributes to poor outcome and this conclusion is justified by the supporting data provided. However, the authors refer to HMOX1 as a transcription factor, but I am not aware of any other literature suggesting that HMOX1/Ho-1 acts as a transcription factor and this work has not shown enough supporting evidence of such. I would suggest updating this language. The raw data has been provided however, there seems to be one discrepancy. In Figure 10 the HMOX1 blot is labeled differently as the raw blots included in the supplementary files/raw data files. Please ensure that this labeling is corrected and that all figures are labeled according to the raw data provided.

Additional comments

The TCGA-HNSC data is categorized by primary site. It may be worth mentioning if there were or were not any significant differences in FTH1 expression based on primary site location.

---

## Round 0.2 · accepted · Accept

The original submission had been examined by two referees. Only one of them reviewed the revised submission. All concerns that were raised in the first review have been satisfactorily addressed with the revisions (none of which involved the presentation or generation of new data).

·

Basic reporting

no comment

Experimental design

no comment

Validity of the findings

no comment

Additional comments

changes to manuscript are sufficient